# Risk of retinal detachment and exposure to fluoroquinolones, common antibiotics, and febrile illness using a self-controlled case series study design: Retrospective analyses of three large healthcare databases in the US

**Ajit A. Londhe**[1¤]**, Chantal E. Holy**[2]*****, James Weaver**[1]**, Sergio Fonseca**[1]**, Angelina Villasis-Keever**[1]**, Daniel Fife**[1]

1 Janssen Pharmaceutical Research & Development, LLC, Titusville, NJ, United States of America,
2 Johnson & Johnson, New Brunswick, NJ, United States of America

¤ Current address: Amgen, Los Angeles, CA, United States of America
* choly1@its.jnj.com

**Data Availability Statement:** Commercial affiliations within J&J did not alter our adherence

## Abstract

### Objective

The risk of retinal detachment (RD) following exposure to fluoroquinolone (FQ) has been assessed in multiple studies, however, results have been mixed. This study was designed to estimate the risk of RD following exposure to FQ, other common antibiotics, and febrile illness not treated with antibiotics (FINTA) using a self-controlled case series (SCCS) study design to reduce risk of confounding from unreported patient characteristics.

### Design

Retrospective database analysis–SCCS.

### Setting

Primary and Secondary Care.

### Study population

40,981 patients across 3 US claims databases (IBM® MarketScan® commercial and Medicare databases, Optum Clinformatics).

### Outcome

RD.

### Methods

Exposures included FQ as a class of drugs, amoxicillin, azithromycin, trimethoprim with and without sulfamethoxazole, and FINTA. For the primary analysis, all drug formulations were

to all PLOS ONE policies on sharing data and materials: the data for these analyses were made available to the authors by third-party licenses from IBM MarketScan and Optum, commercial data providers in the US. The authors have a license for analysis of these data. Under the licensing agreement, the authors cannot provide raw data themselves. Other researchers could access the data by purchase through IBM/Optum, and the inclusion criteria specified in the Methods section would allow them to identify the same cohort of patients we used for these analyses. Future researchers may purchase the data using the following: - For the MarketScan databases: https://www.ibm.com/watson-health/products - For the Optum Database: https://www.optum.com/business/solutions/life-sciences/explore-data.html The authors of this research did not have special access privileges to the data that other researchers would not have.

**Funding:** This study was supported by Janssen Research & Development, LLC, who provided financial support for the work in the form of salaries for all the authors, and its commitment to pay the associated publishing fees. Janssen Research & Development, LLC's Standard Operating Procedures require publication of studies that, like this one, concern its products and were intended for publication when they were begun. For the present study, that intention is documented by its registration with ClinicalTrails. gov. The funding organization's Standard Operating Procedures also require that the study undergo a standard internal company review before it is submitted for publication. Internal company review is limited to roles not responsible for sales or marketing functions. Janssen Research & Development, LLC. did not have any additional role in the study design, data collection, analysis, or preparation of the manuscript. The specific roles of the authors are articulated in the 'author contributions' section.

**Competing interests:** At the time of the work, all authors were full-time employees of Janssen Research & Development, LLC, which is the Marketing Authorisation Holder for Levaquin (a fluoroquinolone) in the United States and some other countries. As full-time employees, the authors held stock or stock options. This does not alter the authors' adherence to PLOS ONE policies on sharing data and materials.

included. For the post hoc sensitivity analyses, only oral tablets were included. Risk windows were defined as exposure period (or FINTA duration) plus 30 days. Patients of all ages with RD and exposures in 3 US claims databases between 2012 to 2017 were included. Diagnostics included p value calibration and pre-exposure outcome analyses. Incidence rate ratios (IRR) and 95% confidence interval (CI) comparing risk window time with other time were calculated.

## Results

Our primary analysis showed an increased risk for RD in the 30 days prior to exposure to FQ or trimethoprim without sulfamethoxazole. This risk decreased but remained elevated for 30 days following first exposure. Our post-hoc analysis, which excluded ophthalmic drops, showed no increased risk for RD at any time, with FQ and other antibiotics.

## Conclusion

Our results did not suggest an association between FQ and RD. Oral FQ was not associated with an increased risk for RD during the pre- or post-exposure period.

## Trial registration

**ClinicalTrials.gov identifier:** NCT03479736-March 21, 2018.

## Introduction

Retinal detachment occurs when the retina separates from the underlying tissue (choroid), which provides oxygen and nutrition to support normal functioning. When separated from the choroid for more than short durations, the photoreceptors within the retina sustain irreversible damage [1]. Retinal detachment affects 10 to 18 people per 100,000 US residents per year, with higher rates among white and older populations [2]. Untreated, retinal detachment can result in complete blindness of the affected eye.

The main tissue responsible for the attachment of the retina to the choroid is collagen. Studies have shown that multiple collagen types are involved in vitreoretinal attachment and that several layers of various types of collagen are found in the retina itself [3]. The interest in fluoroquinolones (FQ) as related to retinal detachment stems from the importance of collagen for retinal attachment and from preclinical study findings associating FQ exposure with impairment of quantity and quality of collagen production [4, 5]. The results of these animal studies have raised ocular safety concerns for patients treated with FQ.

The potential association between RD and FQ exposure has been analyzed in multiple studies with mixed results. Of the most recent 10 studies, 6 reported an association [6–11], and 4 reported no association [12–15]. Two distinct studies used the same database but reported different results [11, 12].

The previously-published analyses include longitudinal population-based cohort studies [7, 9, 13, 15], case control studies [8, 12, 14], a case cross-over study [10], and a sequence symmetry analysis study [6]. The findings were not associated with study design; prospective cohort, case-control, and self-controlled case series (SCCS) results did and did not show associations between RD and FQ exposure.

Some studies included other antibiotics as control exposures and reported no increased risk for RD following FQ exposure relative to exposure beta-lactam [13, 15] and macrolide class antibiotics [15]. Two studies used amoxicillin as a control and reported mixed results, one study suggesting an increased risk for RD following FQ exposure versus amoxicillin [9], the other, no difference in risk [7]. Most studies discussed bias and confounding as key limitations of the findings.

One of the studies that observed an elevated risk for RD after FQ exposure also found an elevated risk for RD before FQ exposure, which led the authors to conclude that "the association between FQ use and retinal detachment might not be a causal relationship" [11]. The reason why findings are not consistent across studies is unclear, however confounding and residual bias may be a key reason for conflicting findings.

To better address confounding, a major limitation of these previous studies, we designed a study that included statistic diagnostics intended to assess and address potential biases:

1. We designed a self-controlled case series (SCCS) study to compare the RD incidence rate between FQ exposed and unexposed time periods within the same patients. The SCCS design protects against confounding by individual characteristics that may be poorly captured in claims databases, e.g., smoking, obesity, and lifestyle, that are known risk factors for RD and that may differ between patient groups in comparative study designs. This design also provides some protection against other biases, such as confounding by indication and protopathic biases, because it allows evaluation of risk both before and after exposures.

2. We also estimated risk of RD following exposure to other commonly prescribed antibiotics (amoxicillin, azithromycin, trimethoprim with and without sulfamethoxazole) and febrile illness not treated with antibiotics (FINTA). These analyses were conducted to contextualize the results from the FQ analysis.

3. We included a diagnostic method to assess residual bias inherent to observational research. Residual bias can occur in large databases and skew results in even well designed studies. Negative controls–exposures known to not cause the outcome of interest–have been used in many studies to evaluate potential residual bias. Calibrating p values is a method that consists of using a large number of negative controls, evaluating their association with the outcome of interest, and defining a p value specific to the given outcome, such that it reflects the actual probability of negative controls yielding a 5% chance of false positive [16–18]. For example, if the negative controls are not centered on the null value or are more scattered than expected, the calibrated p value is able to adjust for this condition. In our study, we used 38 negative controls to evaluate residual bias and estimate calibrated p values, thus establishing a more realistic measure of statistical significance [18].

4. As a second diagnostic method, we also evaluated the risk of RD in time period immediately preceding exposure to FQ, commonly prescribed antibiotics, and FINTA.

5. Finally, to increase the generalizability of our findings, our study was executed against 3 large US claims databases.

Our study was designed to evaluate the association between FQ and three distinct collagen-related adverse events: aortic aneurism or detachment, retinal detachment and Achilles tendon rupture. This current paper focuses on the results of the RD analyses. Results related to aortic aneurism or detachment will be detailed in a separate publication. The analyses for Achilles tendon rupture could not be completed due to excess systematic error.

## Methods

All databases in this study only contained de-identified patient data. No IRB approval was required.

### Data sources

The study was pre-registered on clinicaltrials.gov as NCT03479736. The following databases were used: IBM MarketScan® Commercial Database (IBMCOM), IBM Medicare® Supplemental Database (IBMMDCR), and Optum's De-identified ClinFormatics DataMart—Date of Death (OPTUMEXTDOD). The IBMCOM and IBMMDCR databases include patients with private insurance and together represent 147 million lives. The OPTUMEXTDOD database is also a US administrative health claims database covering 82 million lives. The major data elements contained within these databases are outpatient pharmacy dispensing claims (coded with National Drug Codes (NDC) as well as comprehensive listing of all inpatient and outpatient medical claims with procedure (coded in CPT-4, HCPCs, ICD-9-CM or ICD-10-PCS) and diagnosis codes (coded in ICD-9-CM or ICD-10-CM). The pharmacy dispensing claims contain detailed information on all filled prescriptions. The indication for prescribed drugs, however, is not included in the claims database as this information is not routinely provided on prescriptions.

### Outcome definition

The following criteria were used to define an RD event and were consistent with prior definitions of RD [8, 14]. An RD event required presence of a diagnosis code for RD and an RD procedure record (sclera buckle, vitrectomy, retinopexy, retinal cryotherapy, silicone oil fill, air gas fluid exchange, or pneumatic retinopexy) between 14 days prior to or after first RD diagnosis. The index was defined as the earlier date of RD diagnosis or procedure. Only the first RD event was included in the study.

### Exposures and risk window

The study evaluated the association of the following exposures on risk of RD: all classes of FQ (i.e., ciprofloxacin, gatifloxacin, levofloxacin, norfloxacin, moxifloxacin, gemifloxacin, or ofloxacin), amoxicillin, azithromycin, sulfamethoxazole with trimethoprim, sulfamethoxazole without trimethoprim, and FINTA. For FQ, amoxicillin, azithromycin, and sulfamethoxazole with or without trimethoprim: an exposure period was defined as the number of days of consecutive dispensing of drugs with no interruption of more than 30 days, starting with the date of first dispensing and ending with the end of the days' supply of the last dispensing. All drugs were identified by RxNorm codes for ingredient. The primary analysis included all formulation whereas the post-hoc sensitivity analysis included only tablets, thus excluding ophthalmic drops and injectable formulations. Ophthalmic drops are frequently used prophylactically in patients undergoing eye surgery, even retinal detachment repair surgery because compounds from such drops do not cross the blood retinal barrier and these are therefore considered safe. The use of these drops concurrent with eye surgery was expected to represent a time-varying confounder. Ophthalmic drops also represented nearly half of all FQ prescriptions. This post-hoc analysis was therefore designed to address potential confounding associated with the use of antibiotic-containing ophthalmic drops for patients with RD.

FINTA was defined by a diagnosis of viral disease with fever on the same day, with no antibiotic prescriptions, and no inpatient admissions in the 60 days before and 60 days after the

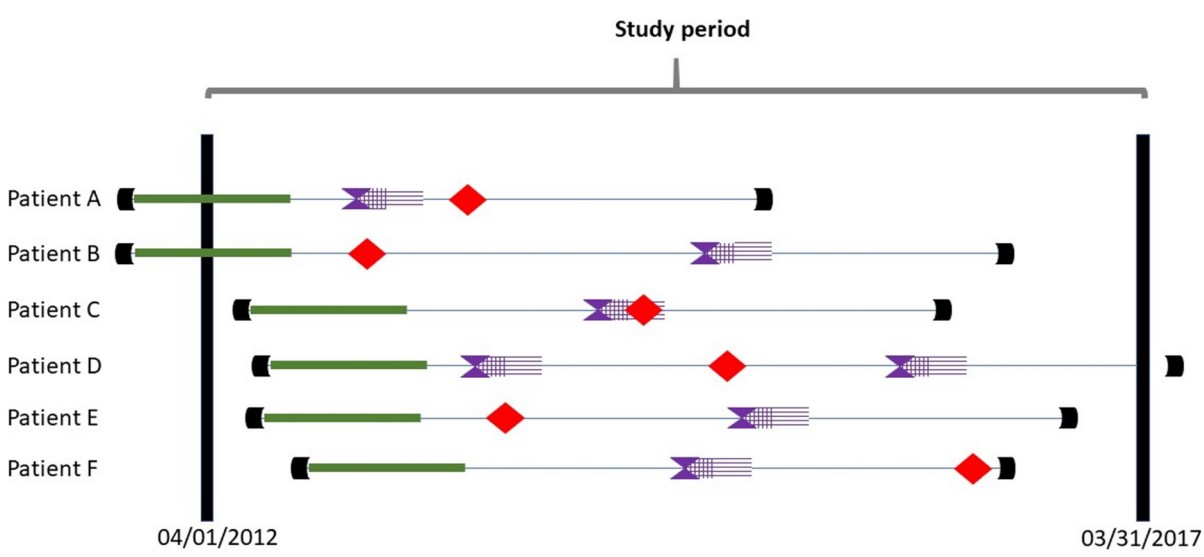

**Fig 1. Graphical representation of all time intervals described in this study.** For example: patient A had a baseline observation period that began approximately 6 months before 01/01/2012, a FQ exposure after the end of the baseline observation period, a retinal detachment after the end of the risk window for that exposure and left the study before 03/31/2017. Patient C had a baseline observation period that began after 04/01/2012, a FQ exposure after the end of the baseline observation period, a retinal detachment within the risk period for that exposure and left the study before 03/31/2017. Patient D had a baseline observation period that began after 04/01/2012, a FQ exposure after the end of the baseline observation period, a retinal detachment after the end of the risk period for that exposure, second exposure after the retinal detachment and ceased to have observations at the end of the study period, i.e., 03/31/2017.

diagnosis. The exposure period was the number of consecutive days with diagnoses of febrile illness. For all exposures (antibiotics and FINTA), the risk window for the primary analysis was defined as the exposure period plus 30 days. The 30-day duration for the risk window was based on Daneman et al, who reported an association with that time window and explained that the 30-day risk window was based on post-marketing surveillance reports that identified tendon complications within 18±24.5 days of FQ exposure [7]. Fig 1 provides a graphical representation of the different time intervals used in this study.

## Study population

The study period ranged from April 1st, 2012 to March 30th, 2017. Patients with at least 1 RD event and 1 exposure of interest and complete risk window during the study period and having at least 12 months of continuous medical and pharmacy benefit enrollment prior to the RD event were initially considered for inclusion in the study. The first 12 months for each patient were considered the naïve period. The end-date of March 30th, 2017 was determined based on the most recent available data at the time of the analysis. The study period start date of April 1st, 2012 ensured at least 5 years of data and considered that other FQ studies on the risk of collagen-related adverse events in the same or similar databases covered data until March 30th, 2012 [14].

## Patient exclusion criteria

Patients were excluded if they met any of the following exclusion criteria: 1) patients who experienced an RD event during a risk window for multiple study exposures of interest (ie multiple different types of antibiotics); these patients were excluded because a definite association with a single study exposure could not be assessed. 2) Patients with the following inherited connective tissue disorders prior to the RD diagnosis: Ehlers-Danlos syndrome, epidermolysis bullosa, Marfan syndrome, or osteogenesis imperfect. 3) Patients with a collagen-related adverse event (RD, Achilles tendon rupture, or aortic aneurism / aortic detachment) in the naïve period. 4) Patients with cataract surgery, iridotomy or iridectomy, eye injury or endophthalmitis or retinitis during the 12-months prior to RD event.

## Sensitivity and post-hoc analyses

Sensitivity analyses included: 1) increasing the risk window to exposure period plus 60 days, and 2) modifying the definition of the RD to further require at least 1 visit with an ophthalmologist at any time prior to the first RD diagnosis to increase confidence that the first RD was indeed the first event given that all patients had access to specialty ophthalmology care. 3) A pre-exposure analysis was also conducted to evaluate impact of possible undefined confounders. For this analysis, pre-exposure IRR were included in the model as covariates. A post-hoc analysis was conducted as described earlier that repeated the entire study with the following new definitions for exposures: for all antibiotics (including FQ), exposures were limited to prescriptions of oral tablets only (thus excluding drops or injection or other formulations), for FINTA, diagnosis of viral disease with fever with no oral tablet antibiotic prescription.

## Sample size assessments

Sample size assessment was performed before the analysis using the method described by Musonda et al. that assesses sample size for an analysis that is done with empirically calibrated confidence intervals. Such analyses, that take account of both random and systematic error, are described below under 'Calibration of p values'; p-values can be larger or smaller than classical p-values and the confidence intervals can be wider or narrower [19].

## Statistical analyses

Incidence rate ratios (IRR) and associated 95% confidence intervals (CI) were calculated to estimate the risk of RD during risk windows and non-risk windows using a Poisson regression conditioned on the event. Analyses were conducted using open-source source software developed in R and SQL [20]. To account for confounding by temporal factors that vary by age and season, linear combinations of cubic splines were modeled to approximate the age and season

effect and adjust the incidence rate ratio accordingly. Given the study population age (>50 years at event date on average) and a 5-year study window, 3 age knots were specified. Five seasonality knots were specified to represent 4 seasons. Empirically calibrated p-values were generated to account for both random and systematic error, as described below.

## Calibration of p values

To estimate residual error in each analysis, 38 exposures known to have no causal association with RD were identified as negative controls [21]. These included: cyclobenzaprine, tramadol, benzonatate, pseudoephedrine, benzoyl peroxide, clobetasol, phenazopyridine, olopatadine, ascorbic acid, fluocinonide, antipyrine, dicyclomine, cefprozil, magnesium sulfate, terbinafine, terconazole, niacin, diphenoxylate, alendronate, permethrin, cetirizine, eszopiclone, oxybutynin, thiamine, phenobarbital, calcipotriene, sodium phosphate, acetic acid, pyrilamine, glucagon, exenatide, selenium sulfide, penciclovir, methylene blue, ciclesonide, clidinium, rifaximin, and loperamide. These negative control exposures were used to calibrate p-values for the association between exposures of interest and RD using the following methodology: the association between each negative control and RD was estimated using the SCCS method and a distribution of those estimates was generated. In absence of bias, negative controls in theory should produce effect estimates with 95% of the associated CIs encompassing 1.0. The observed distribution of the effect estimates of the 38 negative controls was defined as the empirical null distribution, which is interpreted as reflecting the random and systematic error of the study design as applied to the observational database. The empirical null distribution was then used to calibrate p values to reflect the observed systematic error and statistical variability of the estimates, using methodologies described elsewhere [18].

## Pre-exposure analysis

The frequency of RD events was plotted from 60 days prior to up to 60 days after the first day of the first exposure. This analysis provided an evaluation of the distribution of events relative to first exposure, both before and after. In addition, the IRRs during two time periods before first day of first exposure (- 60 to -30-day and -29 to -1-day) were estimated. The IRR from pre-exposure periods were then used as covariates in the sensitivity analyses to further evaluate the impact of time-varying confounders.

## Concurrent drug analysis

The concurrent drugs analysis was similar to the primary analysis. However, for each exposure of interest, all other concurrent drug exposures were included in a regularized conditional Poisson regression to account for the potentially time-varying, confounding effects of multiple drug exposures and, by proxy, the conditions they treat [22].

## Results

Baseline Demographics: Across all 3 databases, a total of 40,981 distinct patients with RD events were identified during the study period. Demographics and exposures for all patients are shown below in Table 1.

## Empirical calibration

Across all databases, analyses using negative control exposures produced effect estimates for RD that were distributed evenly around the hypothetical null (i.e., IRR = 1), suggesting minimal bias as shown in the Supplemental Files (S1 Fig in S2 File). The empirical null distributions

**Table 1. Characteristics and exposure case count for patients with RD.**

| Parameters | OPTUMEXTDOD | IBMMDCR | IBMCOM |
|---|---|---|---|
| RD cases (N) | 13,654 | 4,214 | 23,113 |
| Mean age (years) | 59 | 72 | 51 |
| Female (%) | 41 | 42 | 43 |
| Observation time in years (median) | 5.5 | 5 | 5 |
| Observation time in years (IQR) | 4.96 | 3.64 | 5.89 |
| Cases with FQ class exposure, N(%) | 6,696 (49.04%) | 2,444 (58%) | 9,204 (39.82%) |
| Cases with FINTA exposure, N(%) | 271 (1.98%) | 34 (0.81%) | 469 (2.03%) |
| Cases with Amoxicillin exposure, N(%) | 4,271 (31.28%) | 1,468 (34.84%) | 7,716 (33.38%) |
| Cases with Azithromycin exposure, N(%) | 3,450 (25.27%) | 1,189 (28.22%) | 6,382 (27.61%) |
| Cases with Trimethoprim without Sulfamethoxazole exposure, N(%) | 1,112 (8.14%) | 371 (8.8%) | 2,220 (9.6%) |
| Cases with Trimethoprim with Sulfamethoxazole exposure, N(%) | 1,301 (9.53%) | 522 (12.39%) | 2,705 (11.7%) |

showed moderate random variability. Calibrated p values were calculated and reported in primary and post hoc analyses.

## Exposure timeline

The timeline of RD events, by exposure type, across all 3 databases was plotted. For FQ and trimethoprim without sulfamethoxazole: a peak in RD events occurred just prior to first exposure. That peak decreased but remained greater than the null in the 1–30 days after exposure and decreased to baseline levels by 30 days post first exposure. This finding was observed in all 3 databases. Fig 2 shows the plots specific to FQ exposure. The plot presents the count of RD events (frequency of RD events: y-axis) over the 120-day period centered on the day of first FQ exposure (x = 0). For all other antibiotics (amoxicillin, azithromycin, trimethoprim with sulfamethoxazole), an even distribution of events was observed throughout the 120-day time period. This analysis was repeated for the post-hoc sensitivity analysis that included oral tablets only (and thus excluded ophthalmic drops). For all antibiotics including FQ, events were evenly distributed across the 120-day period, without any peak or noticeable trend. Fig 3 shows the plots specific to the FQ, tablet-only exposures.

The IRR in the pre-exposure intervals (-60 to -30 and -29 to -1 day) were calculated in each database, for all exposure types. Complete tables with all pre-exposure IRRs by exposure type are included in the Supplemental Files (S1-S6 Tables in S2 File). For FQ: IRRs when all formulations of FQ were included were elevated prior to exposure and were highest during the -29 to -1-day interval, ranging from 7.67 (95%CI: 6.94–8.48) in the IBMMDCR database to 12.00 (95%CI: 11.41–12.61) in the IBMCOM database.

When only tablets were included as exposures (post-hoc analysis), there was no increase in IRR in the pre-exposure period. The IRR in the -29 to -1 day ranged from 0.727 (93%CI: 0.595–0.878) in OPTUMEXTDOD to 0.891 (0.771–1.024) in IBMCOM. Trimethoprim without sulfamethoxazole showed a similar trend, with a very high increase in IRR for RD preexposure when all formulations were included, but no increase when only tablets were included. For all other exposures, there was no increased IRR for RD pre-exposure, regardless of formulation.

## Estimate of IRR for RD

Estimates of IRR with 95% confidence intervals and calibrated p values for RD following exposures to FQ, other antibiotics, and FINTA, in each database, are shown in Fig 4 and in the

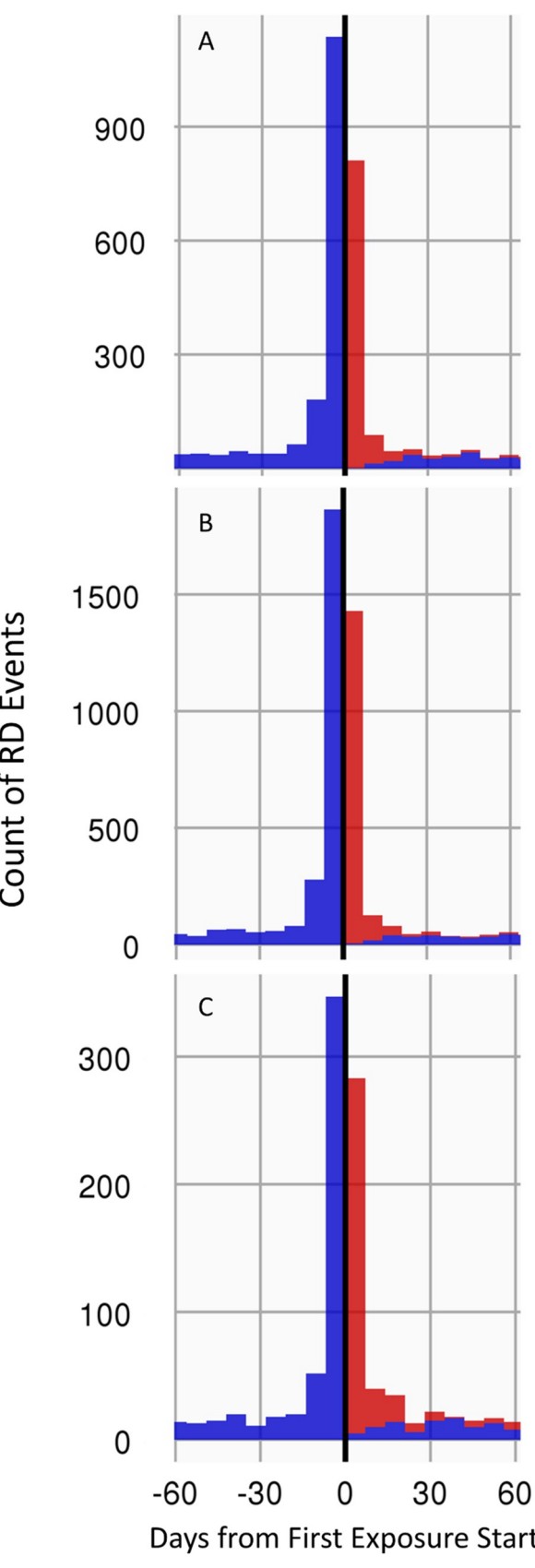

**Fig 2. Timeline of RD Events, in each database, from 60 days before to 60 days after first FQ exposure.** FQ exposure included all types of formulations. Red bars indicate that events occurred concurrently with the FQ risk window, whereas blue bars indicate event occurrences outside of the risk window. A spike of RD events was observed prior to the first day of FQ exposure. 2A: Exposure timeline in OPTUMEXTDOD; 2B: Exposure timeline in IBMCOM; 2C: Exposure timeline in IBMMDCR.

Supplemental Files (S7 and S8 Tables in S2 File). When all formulations were included as exposures, the IRR for RD was significantly elevated for FQ (IRR in: OptumExtDOD: 3.03 (95%CI: 2.84–3.23) IBMMDCR: 3.38 (95%CI: 3.06–3.74) IBMCOM: 3.70 (95%CI: 3.51–3.90)) and for trimethoprim without sulfamethoxazole (IRR in: OptumExtDOD: 4.74 (95%CI: 4.10–5.45); IBMMDCR: 5.19 (4.04–6.62); IBMCOM 5.27 (95%CI: 4.77–5.83)). None of the other exposures were significantly elevated for RD. When exposures included tablets only, none of antibiotics had an IRR significantly elevated for RD. The only elevated risk was with FINTA in the IBMMDCR database (1.83 (95%CI: 1.28–2.55)).

Results from the sensitivity analyses are included in the Supplementary Files (S9-S14 Tables in S2 File) and did not change the conclusions from the primary and post-hoc analyses shown herein.

## Discussion

Our study was designed to evaluate the risk of RD following exposure to FQ as well as amoxicillin, azithromycin, trimethoprim with and without sulfamethoxazole, and FINTA using a SCCS study design that reduces confounding from unobserved patient characteristics. Multiple studies have been conducted to assess associations between FQ exposure and RD, with mixed results. Two prior studies evaluating this association used the SCCS design and reported contradictory results [11, 14]. We therefore conducted two important diagnostic assessments and one post-hoc analysis to address potential biases and better contextualize previously published results. Overall, the contributions of this study versus prior analyses are: 1) the use of a study design that adjusts for time-invariant confounders that are unmeasured or poorly measured in claims databases, and 2) the use of two important diagnostic assessments that were not used in previous studies to evaluate residual bias and timing of events.

To protect against residual error inherent to observational studies, we calibrated p-values to reflect the distribution of estimates obtained using negative control exposures. In our study, negative control estimates distributed evenly around the null, suggesting minimal bias, although the distribution showed moderate random variability. We nevertheless calibrated p values to address the observed residual error to increase confidence in the validity of our findings. It is important to note that p values should not be used for causal inferences–multiple publications have described limitations thereof [23]–but in our study, the use of calibrated p value was important to present the extent of calibration required to account for systematic bias. None of the previously published studies used negative controls-based calibration to account for residual random or systematic error.

To assess the reliability and generalizability of our findings, we conducted our analyses in 3 different claims databases, each comprising large, commercially insured, US populations. It is important to note that 2 prior studies that assessed the association between FQ exposure and RD used the same database yet found different results [11, 12]. Repeating the analysis in multiple databases is therefore important to evaluate the generalizability and consistency of findings.

An important analysis in our study was the evaluation of the event occurrence timeline relative to the time of exposure. This analysis was critical to identify pre-exposure bias in the patient's treatment pathway. Interestingly, when all FQ formulations were included in the

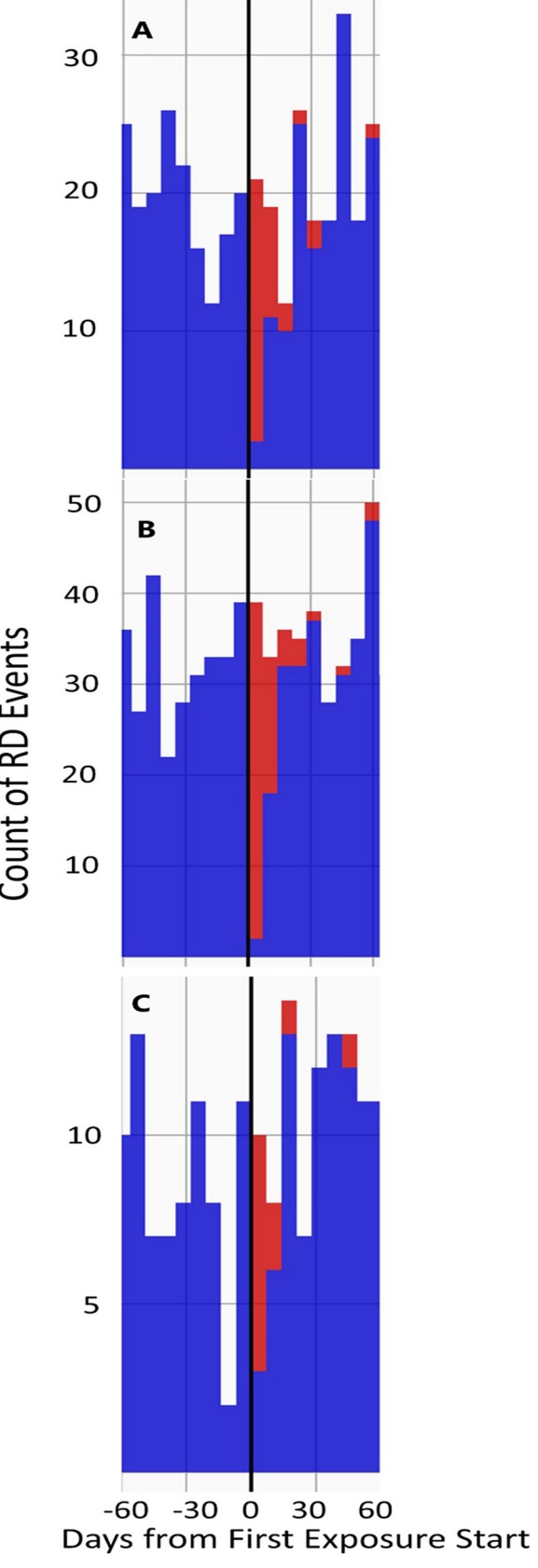

**Fig 3. Timeline of RD Events, in each database, from 60 days before to 60 days after first FQ exposure.** FQ exposure included oral tablets only. Red bars indicate that events occurred concurrently with the FQ risk window, whereas blue bars indicate event occurrences outside of the risk window. 3A: Exposure timeline in OPTUMEXTDOD; 3B: Exposure timeline in IBMCOM; 3C: Exposure timeline in IBMMDCR. In contrast to Fig 2, there is no exposure peak in the days just before the retinal detachment. RD events were randomly distributed before and after the first date of FQ exposure.

exposure definition, we observed that the frequency of RD events increased considerably in the interval ranging from -29 to -1-day before exposure. RD event frequency decreased as time progressed but was still elevated in the 0-30-day interval post FQ exposure. These observations were supported by elevated IRRs during those intervals. A similar finding was recently reported by Shin et al, who observed increased risks for RD in the 30 days following FQ exposure (IRR 1.85; 95% CI 1.71–1.95) and similarly high rate in the 30-day period immediately prior to FQ exposure (IRR 1.58; 95% CI 1.49–1.68) [11]. To our knowledge, none of the other studies evaluated periods preceding exposure. This is however a critical step as pre-exposure bias may thereby be identified, thus further reducing the possibility of a causal association between outcome and exposure.

Our primary analysis assessed risk of RD following exposure to all formulations of FQ and other antibiotics. A close analysis of FQ exposures in our study revealed that ophthalmic drops accounted for 44% of all FQ exposures. Topical ocular medications such as ophthalmic drops do not reach the posterior segment (retina, vitreous, choroid) because they do not cross the anterior segment of the blood-retinal barrier [24]. These medications are therefore unlikely to have a causal association with RD through a direct biochemical mechanism of action. After ocular surgery, patients in the United States are routinely prescribed prophylactic antibiotics to prevent endophthalmitis, and most prescribed antibiotics are ophthalmic drugs. This fact explains why so many of our FQ exposures were in fact ophthalmic drops, but also highlighted the need to repeat our analysis with non-ophthalmic formulations of FQ.

Our post-hoc analysis was therefore designed to only focus on oral FQ in order to explicitly exclude exposures that 1) may result from ocular treatments, which are potential indicators of reverse causality, and 2) are unlikely to cause RD because these formulations do not reach the retina. As shown in Fig 4, systemic, oral FQ formulations showed no obvious temporal association with RD events, the vast majority of RD cases were not observed in this analysis. Because this post-hoc analysis addressed the unanticipated time-varying confounder that resulted from including ophthalmic drops in the exposure definition of the pre-planned primary analysis, it is our post-hoc sensitivity analysis–and not the primary analysis—that yielded correct and reliable estimates.

Our analysis therefore identified findings consistent with Shin et al, suggesting reverse causality between RD events and FQ exposure, and more importantly identified no association when FQ oral formulations were analyzed separately from all FQ formulations [11].

Interestingly, the IRR associated with FINTA exposure changed slightly between the primary and the post-hoc analysis. The explanation for this change is as follows: in the primary analysis, "FINTA" was defined as fever with no antibiotic, regardless of what formulation of antibiotic. In the post-hoc sensitivity analysis, antibiotics included tablets only, and FINTA could therefore include exposures during which patients experienced fever untreated with (oral) antibiotics but with antibiotics in the form of ophthalmic drops. This change in definition resulted in a very small change in the FINTA exposure risk between the primary and the post-hoc sensitivity analysis.

There are several key limitations to our study. The SCCS study design requires that the probability of exposure be independent of occurrence of event. This requirement is difficult to

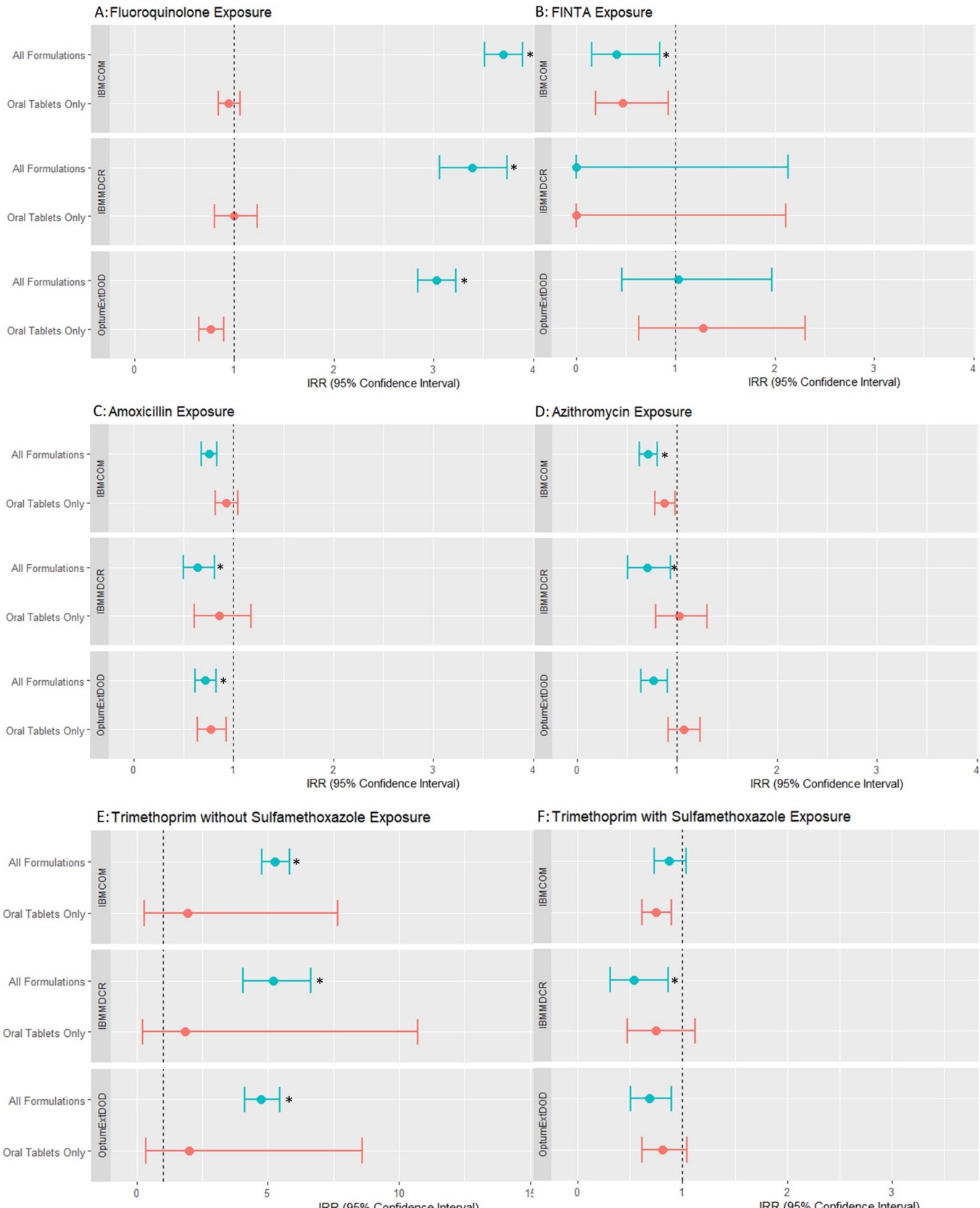

**Fig 4. Incidence risk ratios (IRR) by exposure and database.** Asterisks identify statistically meaningful findings based on calibrated p values. Risks associated with exposures that included tablets only are shown in blue lines whereas risks associated with exposures that included all formulations are shown in orange lines. For fluoroquinolone exposures and for trimethoprim without sulfamethoxazole exposures, there is a substantial and statistically significant association of RD with exposure across all databases when all formulations, including topical ophthalmic formulations are included. When only oral exposures are included, the magnitude of this this association for these two medications becomes much smaller across all databases and the association loses statistical significance except in the panel for trimethoprim without sulfamethoxazole in the IBMMCDR database.

ascertain in the context of antibiotic use and events that may possibly include prophylactic antibiotic prescriptions. In our study, a few steps were taken to reduce this risk. First, only the first RD event for any patient was included in the study, thus precluding an RD in the second eye, with its increased or decreased risk vs baseline. Events occurring during risk windows for multiple exposures were also excluded, thus eliminating cases where definite associations could not be drawn. Finally, a large pre-event peak in exposure could be due to prophylactic treatment of patients following eye surgery, such that exposure–if that hypothesis is true–may not be independent of event. To address this potential limitation, our post hoc analysis excluding ophthalmic drops was conducted. In this scenario, no peak was observed. An additional potential bias may be related to prescriber awareness of potential detrimental effect of fluoroquinolones on retinal detachment, thus affecting the independence between exposure and events. The use of negative controls for p value calibration, while described in multiple papers, is not currently considered standard and there is ongoing debate on the most appropriate approaches to evaluate significance [16, 18, 25, 26]. In addition, the datasets used in this analyses (IBM and Optum) may have overlapping patients, complete independence between the databases cannot be ascertained. An additional limitation may be related to unobserved confounders: whereas the SCCS study design allows for good control over fixed confounders, it does not eliminate the potential for all time-varying confounders. Our study used prescription information from claims databases. These databases capture prescriptions as they are filled–and partially paid for–by patients, but there is no certainty that patients actually took the drugs after filling the prescriptions. Finally, our study may be subject to additional limitations that affect studies based on secondary data.

In conclusion, our analyses did not confirm prior studies that report an increased risk of RD following exposure to FQ. From the perspective of hypothesis testing, the association was not statistically significant; from the perspective of estimation, the upper endpoints of the 95% confidence intervals were close to the null [27]. A peak in RD events was observed in the 30 days prior to the first FQ exposure when all formulations of FQ were included in the analysis, which suggests that factors preceding FQ exposure are responsible for the events. When oral-only FQ formulations were analyzed, no association was observed at either shortly prior to or after FQ exposure.

## Supporting information

**S1 File.**
(PDF)

**S2 File. Exposure timeline and sensitivity analyses.** Negative Control Diagnostics and Incidence Rate Ratio Estimates for Retinal Detachment in all Databases, for Primary, Sensitivity and Post-hoc Analyses.
(RTF)

## Acknowledgments

The authors would like to acknowledge Dr. Li Lin for editorial writing support, and Ms. Prerna Kothari for programming support.

## Author Contributions

**Conceptualization:** Chantal E. Holy, Sergio Fonseca, Angelina Villasis-Keever, Daniel Fife.

**Data curation:** Ajit A. Londhe, James Weaver.

**Formal analysis:** Ajit A. Londhe, James Weaver.

**Investigation:** Sergio Fonseca, Angelina Villasis-Keever, Daniel Fife.

**Methodology:** Ajit A. Londhe, Chantal E. Holy, James Weaver, Sergio Fonseca, Angelina Villasis-Keever, Daniel Fife.

**Project administration:** Chantal E. Holy, Sergio Fonseca, Daniel Fife.

**Resources:** Chantal E. Holy, Sergio Fonseca.

**Software:** Ajit A. Londhe, James Weaver.

**Supervision:** Chantal E. Holy, Sergio Fonseca, Angelina Villasis-Keever, Daniel Fife.

**Validation:** Ajit A. Londhe, James Weaver, Angelina Villasis-Keever.

**Visualization:** Ajit A. Londhe, James Weaver.

**Writing – original draft:** Chantal E. Holy, James Weaver, Sergio Fonseca, Angelina Villasis-Keever.

**Writing – review & editing:** Chantal E. Holy, Daniel Fife.

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
