## [Decision Letter · Decision Letter 0]

9 Jun 2022

PONE-D-21-26171Risk of Retinal Detachment and Exposure to Fluoroquinolones, Common Antibiotics, and Febrile Illness Using a Self-Controlled Case Series Study Design: Retrospective Analyses of Three Large Healthcare Databases in the US.PLOS ONE

Dear Dr. Holy,

Thank you for submitting your manuscript to PLOS ONE. Firstly, we would like to apologize for the delay in processing your manuscript. It has been exceptionally difficult to secure reviewers to evaluate your study. We have now received one completed review, which is available below. The reviewer has raised concerns about the study that need to be addressed in a revision.

Please note that we have only been able to secure a single reviewer to assess your manuscript. We are issuing a decision on your manuscript at this point to prevent further delays in the evaluation of your manuscript. Although the reviewer is quite positive about this study, it has been noted that only the statistical part of your work has been assessed. Please be aware that the editor who handles your revised manuscript might find it necessary to invite additional reviewers to assess this work once the revised manuscript is submitted. However, we will aim to proceed on the basis of this single review if possible.

After careful consideration, we feel that it has merit but does not fully meet PLOS ONE’s publication criteria as it currently stands. Therefore, we invite you to submit a revised version of the manuscript that addresses the points raised during the review process.

We look forward to receiving your revised manuscript.

Kind regards,

Miquel Vall-llosera Camps

Senior Editor

PLOS ONE

Journal Requirements:

Additional Editor Comments:

We noted that this manuscript is closely related to previously published works, including: "Risk of aortic aneurysm and dissection following exposure to fluoroquinolones, common antibiotics, and febrile illness using a self-controlled case series study design: Retrospective analyses of three large healthcare databases in the US (https://doi.org/10.1371/journal.pone.0255887)". 

As outlined in our publication criteria, we discourage the unnecessary division of work into parts and we ask that sufficient rationale be given for submissions that are derivative of or closely related to previously work.

Please provide a clear rationale for the necessity of separating this related research in different manuscripts. We would expect clarification on the differences in the research questions addressed and the data sets used in the two manuscripts, and confirm that the separation into more than one article has not compromised the robustness of the statistical analysis.

Reviewers' comments:

Reviewer's Responses to Questions

**Comments to the Author**

1. Is the manuscript technically sound, and do the data support the conclusions?

Reviewer #1: Yes

2. Has the statistical analysis been performed appropriately and rigorously? 

Reviewer #1: Yes

3. Have the authors made all data underlying the findings in their manuscript fully available?

Reviewer #1: Yes

4. Is the manuscript presented in an intelligible fashion and written in standard English?

Reviewer #1: Yes

5. Review Comments to the Author

Reviewer #1: Important note: This review pertains only to ‘statistical aspects’ of the study and so ‘clinical aspects’ [like medical importance, relevance of the study, ‘clinical significance and implication(s)’ of the whole study, etc.] are to be evaluated [should be assessed] separately/independently. Further please note that any ‘statistical review’ is generally done under the assumption that (such) study specific methodological [as well as execution] issues are perfectly taken care of by the investigator(s). This review is not an exception to that and so does not cover clinical aspects {however, seldom comments are made only if those issues are intimately / scientifically related & intermingle with ‘statistical aspects’ of the study}. Agreed that ‘statistical methods’ are used as just tools here, however, they are vital part of methodology [and so should be given due importance].

COMMENTS: I must say that this study is excellently conducted. Manuscript is also nicely drafted. Though it is a “pen & paper” exercise [being a ‘secondary data analyses’], hard work of the authors can be seen and is highly appreciated. In this article, I believe that, figures are very important (to understand the execution), however, in my opinion the explanation (of figures) is inadequate.

One question [sometimes] may be asked like as per line 141 of the manuscript [section on ‘Study Population’] “The study period ranged from April 1st, 2012 to March 30th, 2017”, then what are/is the reason(s) for delay in publication? is nicely answered in the same paragraph by stating that “The end-date of March 30th, 2017 was determined based on the most recent available data at the time of the analysis which ensure at least 5 years required data. {by-and-large people do not like old data analyses}

Since self-controlled case series studies are rarely conducted [probably due to limitations/clumsiness] and sample size is a very important issue, little more in ‘Sample Size Assessments’ section [lines 168-70] {as well more on design - how executed} would have been appreciated.

I highly appreciate a phase/sentence used in line 24 [Conclusion: Our results did not suggest an association between FQ and RD.] because one should always remember that “Absence of evidence is not evidence of absence” [Altman DG, Bland JM. BMJ volume 311, 1995, p 485 (Reprinted: Australian Veterinary Journal 1996;74, 311)]. {Even when P-value is not significantly lower that is null hypothesis of no difference / no association is not rejected, (in short, result is not significant), that does not amount to evidence of absence i.e., it does not imply that there no difference / no association. It only implies that there is no (i.e., these samples do not provide) [say enough] evidence to prove (rather indicate with certain specified confidence level) the difference / association}. This fact is often ignored.

Several limitations of this study as described in lines 339-355. It is true that ‘study used prescription information from claims databases’, however, note that any ‘secondary data analyses’ itself is a great limitation. I recommend acceptance of this article with few minor [these learned authors know where/what to improve] revision.

6. PLOS authors have the option to publish the peer review history of their article (what does this mean?). If published, this will include your full peer review and any attached files.

Reviewer #1: No

---

## [Author Response · Author response to Decision Letter 0]

8 Aug 2022

RESPONSE TO THE COMMENTS: The authors would like to thank the Reviewer for the thorough comments, which have been addressed and resulted in a greatly improved manuscript.

COMMENTS – Manuscript PONE-D-21-26171

Title: “Risk of Retinal Detachment and Exposure to Fluoroquinolones, Common Antibiotics, and Febrile Illness Using a Self-Controlled Case Series Study Design: Retrospective Analyses of Three Large Healthcare Databases in the US.”

Important note: This review pertains only to ‘statistical aspects’ of the study and so ‘clinical aspects’ [like medical importance, relevance of the study, ‘clinical significance and implication(s)’ of the whole study, etc.] are to be evaluated [should be assessed] separately/independently. Further please note that any ‘statistical review’ is generally done under the assumption that (such) study specific methodological [as well as execution] issues are perfectly taken care of by the investigator(s). This review is not an exception to that and so does not cover clinical aspects {however, seldom comments are made only if those issues are intimately / scientifically related & intermingle with ‘statistical aspects’ of the study}. Agreed that ‘statistical methods’ are used as just tools here, however, they are vital part of methodology [and so should be given due importance]. 

COMMENTS: I must say that this study is excellently conducted. Manuscript is also nicely drafted. Though it is a “pen & paper” exercise [being a ‘secondary data analyses’], hard work of the authors can be seen and is highly appreciated. In this article, I believe that, figures are very important (to understand the execution), however, in my opinion the explanation (of figures) is inadequate. 

Author response: All figure captures have been modified to include additional explanations. 

 One question [sometimes] may be asked like as per line 141 of the manuscript [section on ‘Study Population’] “The study period ranged from April 1st, 2012 to March 30th, 2017”, then what are/is the reason(s) for delay in publication? is nicely answered in the same paragraph by stating that “The end-date of March 30th, 2017 was determined based on the most recent available data at the time of the analysis which ensure at least 5 years required data. {by-and-large people do not like old data analyses}

Author response: Peer-review of our work was unfortunately delayed due to difficulties outside our control, in particular changes in focus due to COVID-19.

Since self-controlled case series studies are rarely conducted [probably due to limitations/clumsiness] and sample size is a very important issue, little more in ‘Sample Size Assessments’ section [lines 168-70] {as well more on design - how executed} would have been appreciated. 

Author response: Additional comments were added throughout the text, to better explain our approach and findings. 

I highly appreciate a phase/sentence used in line 24 [Conclusion: Our results did not suggest an association between FQ and RD.] because one should always remember that “Absence of evidence is not evidence of absence” [Altman DG, Bland JM. BMJ volume 311, 1995, p 485 (Reprinted: Australian Veterinary Journal 1996;74, 311)]. {Even when P-value is not significantly lower that is null hypothesis of no difference / no association is not rejected, (in short, result is not significant), that does not amount to evidence of absence i.e., it does not imply that there no difference / no association. It only implies that there is no (i.e., these samples do not provide) [say enough] evidence to prove (rather indicate with certain specified confidence level) the difference / association}. This fact is often ignored. 

Author response: The Authors would like to thank the reviewer for reminding us of this excellent reference, which was added to our manuscript.

Several limitations of this study as described in lines 339-355. It is true that ‘study used prescription information from claims databases’, however, note that any ‘secondary data analyses’ itself is a great limitation. I recommend acceptance of this article with few minor [these learned authors know where/what to improve] revision.

Author response: Additional limitations were added to the discussion section.

---

## [Decision Letter · Decision Letter 1]

22 Aug 2022

PONE-D-21-26171R1Risk of Retinal Detachment and Exposure to Fluoroquinolones, Common Antibiotics, and Febrile Illness Using a Self-Controlled Case Series Study Design: Retrospective Analyses of Three Large Healthcare Databases in the US.PLOS ONE

Dear Dr. Holy,

Thank you for submitting your manuscript to PLOS ONE. After careful consideration, we feel that it has merit but does not fully meet PLOS ONE’s publication criteria as it currently stands. Therefore, we invite you to submit a revised version of the manuscript that addresses the points raised during the review process.

We look forward to receiving your revised manuscript.

Kind regards,

Kuo-Cherh Huang

Academic Editor

PLOS ONE

Journal Requirements:

Additional Editor Comment:

Dear Dr. Holy,

Thank you for submitting your revised manuscript to PLoS ONE. Although the revised manuscript is much improved, the need for some revisions remains. Reviewer 2 has provided several critical points to help advance your work; for example, the recommendation of carrying out sensitivity analyses. Please respond to each comment of Reviewer 2 carefully and thoroughly. Please explain where you feel you cannot completely agree with reviewer’s suggestions.

Kuo-Cherh Huang

Reviewers' comments:

Reviewer's Responses to Questions

**Comments to the Author**

1. If the authors have adequately addressed your comments raised in a previous round of review and you feel that this manuscript is now acceptable for publication, you may indicate that here to bypass the “Comments to the Author” section, enter your conflict of interest statement in the “Confidential to Editor” section, and submit your "Accept" recommendation.

Reviewer #1: All comments have been addressed

Reviewer #2: (No Response)

2. Is the manuscript technically sound, and do the data support the conclusions?

Reviewer #1: (No Response)

Reviewer #2: Yes

3. Has the statistical analysis been performed appropriately and rigorously? 

Reviewer #1: (No Response)

Reviewer #2: Yes

4. Have the authors made all data underlying the findings in their manuscript fully available?

Reviewer #1: (No Response)

Reviewer #2: No

5. Is the manuscript presented in an intelligible fashion and written in standard English?

Reviewer #1: (No Response)

Reviewer #2: Yes

6. Review Comments to the Author

Reviewer #1: COMMENTS: Since all of the comments made on earlier draft are considered positively, I recommend the acceptance because the manuscript now has achieved acceptable level, in my opinion.

Reviewer #2: The authors should be commended on their methodologically rigorous approach and well written interpretation of analyses.

I have several comments:

1) Given the unique nature of the SCCS design, I would suggest including several more summary statistics in Table 1 relevant to this design and for each database

i) mean(sd)/median(iqr) of exposures per observation period

ii) mean(sd)/median(iqr) of intervals per observation period

iii) mean(sd)/median(iqr) of interval durations (days)

iv) total duration of observation period (days)

v) ratio of total duration of exposure/non-exposure periods

2) Did the authors ensure that there was at least a 60 day observation period after the final exposure day prior to ending the observation period?

3) The risk of violating the independence assumption between RD and antibiotic exposure (not relevant for FINTA) may be significant, especially for those patients who experienced an RD event in the pre-exposure (-60 to 0 days) period for 2 reasons: i) a RD in the pre-exposure period would intuitively reduce the risk of a similar event in the post-exposure period as a patient would likely have only half the risk (one eye) as opposed to 2 eyes which might be affected by the exposure, and ii) the odds of prescribing a FQ compared to another antibiotic in those with an RD even in the pre-exposure period might be significantly reduced due to pre-existing concerns. While the authors included the pre-exposure IRR as covariates in the final regression analyses, it would be useful to conduct a sensitivity analyses to test for this violation using the following:

1) estimate the odds ratio of prescribing FQ versus other antibiotic exposure in those with pre-exposure RD event versus those without, and

2) estimate the marginal difference in the mean interval length (days) (and 95% CI) between RD-antibiotic exposure events versus antibiotic-antibiotic exposure events

7. PLOS authors have the option to publish the peer review history of their article (what does this mean?). If published, this will include your full peer review and any attached files.

Reviewer #1: **Yes: **Dr. Sanjeev Sarmukaddam

Reviewer #2: No

---

## [Author Response · Author response to Decision Letter 1]

21 Sep 2022

Response to Reviewers 

Reviewer #2: The authors should be commended on their methodologically rigorous approach and well written interpretation of analyses.

Authors Response: Thank you.

I have several comments:

1) Given the unique nature of the SCCS design, I would suggest including several more summary statistics in Table 1 relevant to this design and for each database

i) mean(sd)/median(iqr) of exposures per observation period

ii) mean(sd)/median(iqr) of intervals per observation period

iii) mean(sd)/median(iqr) of interval durations (days)

iv) total duration of observation period (days)

v) ratio of total duration of exposure/non-exposure periods (CHECK CSR)

Author Response: The requested statistics would require considerable re-work and re-analyses, for which we are not resourced. We believe, however, that these requests might be driven by the Reviewer’s interest in the possibility that our negative findings might be due to insufficient power. Prior to the study conduct, we conducted a power calculation, as shown in the Protocol on page 13 section 7: and reported in Sample size and Study Power (Supplemental Files). From our final results, the best estimate of the variability is thus given by the confidence intervals, specified for all exposures. 

2) Did the authors ensure that there was at least a 60 day observation period after the final exposure day prior to ending the observation period?

Author Response: We thank the Reviewer for this question and have added the corresponding detail in the Methods section: all cases had a complete risk period, defined as an exposure period with a buffer of 30- or 60- days (primary vs sensitivity analyses). (Methods: Line 148) 

3) The risk of violating the independence assumption between RD and antibiotic exposure (not relevant for FINTA) may be significant, especially for those patients who experienced an RD event in the pre-exposure (-60 to 0 days) period for 2 reasons: i) a RD in the pre-exposure period would intuitively reduce the risk of a similar event in the post-exposure period as a patient would likely have only half the risk (one eye) as opposed to 2 eyes which might be affected by the exposure, and ii) the odds of prescribing a FQ compared to another antibiotic in those with an RD even in the pre-exposure period might be significantly reduced due to pre-existing concerns. While the authors included the pre-exposure IRR as covariates in the final regression analyses, it would be useful to conduct a sensitivity analyses to test for this violation using the following:

1) estimate the odds ratio of prescribing FQ versus other antibiotic exposure in those with pre-exposure RD event versus those without, and

2) estimate the marginal difference in the mean interval length (days) (and 95% CI) between RD-antibiotic exposure events versus antibiotic-antibiotic exposure events

Author Response: 

- In response to i): This is an important point. As indicated on line 111: only the first RD event was included in the study, and all patients had at least 365 days of naïve period. No patients had RD captured in the database prior to the observation period. A statement was added in the Discussion section to further reinforce this point. (Line 354-357).

- In response to ii): The Authors agree with this concern, and we further strengthened the Discussion on this topic. Cases of RD within risk periods of more than one exposure were not included. An important exclusion criterion for our study (line 156) is: 

“patients who experienced an RD event during a risk window for multiple study exposures of interest (ie multiple different types of antibiotics); these patients were excluded because a definite association with a single study exposure could not be assessed.”

The real world experience of RD occurrence during risk windows of potentially multiple antibiotics has therefore been removed from our analysis, to reach a more unbiased estimate. 

This consideration was also added in the Discussion section, line 354-357.

---

## [Decision Letter · Decision Letter 2]

26 Sep 2022

Risk of Retinal Detachment and Exposure to Fluoroquinolones, Common Antibiotics, and Febrile Illness Using a Self-Controlled Case Series Study Design: Retrospective Analyses of Three Large Healthcare Databases in the US.

PONE-D-21-26171R2

Dear Dr. Holy,

We’re pleased to inform you that your manuscript has been judged scientifically suitable for publication and will be formally accepted for publication once it meets all outstanding technical requirements.

Kind regards,

Kuo-Cherh Huang

Academic Editor

PLOS ONE

Additional Editor Comments (optional):

Reviewers' comments:

Reviewer's Responses to Questions

**Comments to the Author**

1. If the authors have adequately addressed your comments raised in a previous round of review and you feel that this manuscript is now acceptable for publication, you may indicate that here to bypass the “Comments to the Author” section, enter your conflict of interest statement in the “Confidential to Editor” section, and submit your "Accept" recommendation.

Reviewer #2: All comments have been addressed

2. Is the manuscript technically sound, and do the data support the conclusions?

Reviewer #2: (No Response)

3. Has the statistical analysis been performed appropriately and rigorously? 

Reviewer #2: (No Response)

4. Have the authors made all data underlying the findings in their manuscript fully available?

Reviewer #2: (No Response)

5. Is the manuscript presented in an intelligible fashion and written in standard English?

Reviewer #2: (No Response)

6. Review Comments to the Author

Reviewer #2: (No Response)

7. PLOS authors have the option to publish the peer review history of their article (what does this mean?). If published, this will include your full peer review and any attached files.

Reviewer #2: No

---

## [Editor Report · Acceptance letter]

27 Sep 2022

PONE-D-21-26171R2 

 Risk of Retinal Detachment and Exposure to Fluoroquinolones, Common Antibiotics, and Febrile Illness Using a Self-Controlled Case Series Study Design: Retrospective Analyses of Three Large Healthcare Databases in the US. 

Dear Dr. Holy:

I'm pleased to inform you that your manuscript has been deemed suitable for publication in PLOS ONE. Congratulations! Your manuscript is now with our production department. 

Kind regards, 

on behalf of

Dr. Kuo-Cherh Huang 

Academic Editor

PLOS ONE